# Immaturin-Nuclease as a Model System for a Gene-Programmed Sexual Development and Rejuvenescence in *Paramecium* Life History

**DOI:** 10.3390/microorganisms11010082

**Published:** 2022-12-28

**Authors:** Nobuyuki Haga, Toshinori Usui, Yasuhiro Takenaka, Yuta Chiba, Tomoaki Abe

**Affiliations:** 1Department of Biological Sciences, Faculty of Science and Technology, Senshu University of Ishinomaki, Miyagi 986-8580, Japan; 2Department of Physiology, Graduate School of Medicine, Nippon Medical School, Tokyo 113-8602, Japan

**Keywords:** sexual development, microinjection, rejuvenescence, immaturin-nuclease, *Paramecium*

## Abstract

Fertilization-initiated development and adult-onset aging are standard features in the life history of eukaryotes. In *Paramecium*, the number of cell divisions after the birth of a new generation is an essential parameter of sexual phase transition and aging. However, the gene driving this process and its evolutionary origin have not yet been elucidated. Here we report several critical outcomes obtained by molecular genetics, immunofluorescence microscopy, transformation by microinjection, and enzymological analysis. The cloned immaturin gene induces sexual rejuvenation in both mature and senescent cells by microinjection. The immaturin gene originated from proteobacteria’s glutathione-S-transferase (GST) gene. However, immaturin has been shown to lose GST activity and instead acquire nuclease activity. In vitro substrates for immaturin-nuclease are single- and double-stranded DNA, linear and circular DNA, and single-stranded viral genome RNA such as coronavirus. Anti-immaturin antibodies have shown that the subcellular localizations of immaturin are the macronucleus, cytoplasm, cell surface area, and cilia. The phase transition of sexuality is related to a decrease in the intracellular abundance of immaturin. We propose that sexual maturation and rejuvenation is a process programmed by the immaturin gene, and the sexual function of each age is defined by both the abundance and the intracellular localization mode of the immaturin-nuclease.

## 1. Introduction

Some species of ciliates accomplish conjugation processes that are equivalent to the fertilization and early development events of metazoans. In the life history of *Paramecium*, the conjugation process is initiated by the species-specific sexual cell contact between mature cells of complementary mating types. Sexual cell contact is achieved with the cilia located on the ventral surface of cells [1,2]. In most species of ciliates, the length of the immaturity period is determined by the number of cell divisions after conjugation, and this number differs among species [3,4,5,6]. An analysis of a few mutants that have altered the length of immaturity provides compelling evidence that genetic factors influence the duration of the immature phase [7,8,9]. Sonneborn proposed the importance of immaturity based on population genetics; the presence of sexual immaturity provides time for offspring from the same mating to scatter around each other, preventing mating between closest relatives and avoiding inbreeding [3]. The length of the immature period of ciliate species provides an experimental system suitable for studying the phase transition of life history during the development time. In *Paramecium caudatum* (Pc), the immaturity period lasts for about 50 fissions [5]. The transfer of cytoplasm from immature cells to mature cells via microinjection inhibited mating in the injected cells; these experiments led to the identification of a cytoplasmic factor in immature cells that inhibits the mating activity of mature cells under the experimental conditions [10]. Cross-species transfer of cytoplasm was used to investigate the evolution of this cytoplasmic factor among *Paramecium* species. In some cases, cytoplasmic transfer between species inhibited mating activity, indicating functional conservation. In other cases, cytoplasmic transfer between species did not affect mating activity, indicating functional divergence [11,12].

The factor responsible for immaturity was then isolated and characterized as a small heat-labile protein, referred to as immaturin [13]. Immaturin isolated from the cytoplasm of immature Pc plays a central role in the repression of mating activity in mature cells. Figure 1 shows the schematic of the immaturin bioassay using the microinjection and capillary culture method. Interestingly, microinjection of either immature cell cytoplasm or purified immaturin into senescent cells renewed mating activity in recipient senescent cells [14]. Thus, immaturin treatment induced rejuvenation in the life history stage from senescence to maturity. In addition, immature macronuclei have a strong immaturin-like effect when introduced into mature cells. The immature macronucleus suppressed the mating activity of mature cells. In particular, the immature nucleus age was negatively correlated with the magnitude of suppression. In macronucleus-transplantation experiments, it has been observed that macronuclei from immature cells that had undergone more division were less effective than those from younger cells [15]. These findings indicate that the macronucleus acts as a counter for counting cell division and immaturin indicates the number of cell divisions in the generation. However, the genes and physiological functions of immaturin remain elusive. In this paper, we demonstrate the characteristics of the immaturin gene and the physiological functions of immaturin polypeptides. We also hypothesize that both the sexual maturation process and sexual aging are dependent on the action of the immature gene.

## 2. Materials and Methods

### 2.1. Stocks and Cell Culture

We used stocks belonging to *Paramecium caudatum*, Syngen 3 (Symbiosis Laboratory, Yamaguchi Univ., Yamaguchi, Japan) provided the strain TAZ0460. Paramecia were cultured in 1.25% (*w*/*v*) fresh lettuce juice diluted with K-DS (Dryl’s solution modified by substituting KH_2_PO_4_ for NaH_2_PO_4_, pH 7.0 [16]) and inoculated with *Klebsiella pneumoniae* one day before use [1]. All cells were grown at 25 °C.

### 2.2. Microinjection

Koizumi’s method of microinjecting paramecia [17] was modified as described by Haga et al. [18]. Approximately 40 pL of the solution containing immaturin was injected into the cytoplasm of the *Paramecium* cells for the immaturin assay. Approximately 5–10 pL of the plasmid-containing solution was injected into the macronuclei of cells for transformation experiments.

### 2.3. Immaturin Assay

The schematic of the improved immaturin bioassay is summarized in Figure 1. The assay involves the following steps. Step 1: A mature cell was injected with about 40 pL of immaturin-containing solution. Step 2: Each recipient stood in a cell-free culture medium overnight at 25 °C. Step 3: The recipient was transferred into a fresh culture medium. Step 4: The recipient was sucked into a capillary (0.6 mm inside diameter, 75 mm long) by the capillary phenomenon. Step 5: Each capillary culture (about 20 μL of culture medium) was incubated at 25 °C for about four days. The recipient divided for about four fissions, resulting in 16 daughter cells, and reached the stationary phase.

Step 6: All cells were transferred to a depression slide glass, and some were retransferred to a new depression slide glass. Step 7: After counting the cells, about one hundred of the mating-reactive cells of the complementary mating type (tester cells) were added and incubated for 5 min at 25 °C. A mating clump is usually composed of several tester cells for each cell to be tested. The mating clumps were gently transferred into the behavioral test solution (40 mM KCl in Dryl’s solution) with a micropipette. In this operation, only the cells that underwent a true mating reaction in the head-to-head connection remained bound. The other cells dispersed in different directions by swimming backward. The number of mating reactive cells was counted as half the number of cells that led to the head-to-head joining.

### 2.4. Immaturin Inhibition by Anti-Immaturin Antibody

The soluble fraction of immature cells at about 20 fissions after conjugation obtained by ultracentrifugation at 105,000× *g* for 1 h (at 4 °C) was used as the immaturin fraction (I-supernatant, 10.0 mg protein/mL). The I-supernatant fraction was incubated with the anti-immaturin rabbit polyclonal antibody NH3545 (0.31 or 0.15 mg protein/mL) for 24 h at 4 °C and microinjected into mature cells.

### 2.5. Preparation of Recombinant Immaturin and Its Effects

The coding region of the immaturin gene was de novo synthesized (Eurofins genomics) to optimize its codon usage for expression in *E. coli*, and inserted into pET16b (Merck Millipore, Darmstadt, Germany). 10× His-tagged immaturin protein was overexpressed in BL21(DE3) at 37 °C for 24 h using 50 mL auto-induction media, ZYM-5052.

### 2.6. The Maximum Likelihood (ML) Consensus Tree of Immaturin

A phylogenetic tree was generated from bootstrap analysis with 1000 replications of immaturin and orthologs based on the amino acid sequence alignment, as shown in Appendix A.

### 2.7. Immaturin Imaging in Intracellular Localizations: Direct Immunofluorescent Staining

Before the staining procedure, paramecia were allowed to stand in K-DS for at least 30 min to remove food vacuoles from the cells. They were incubated in 0.1% (*v*/*v*) of osmic acid solution for 30 s to fix paramecia. After fixation, the cells were washed twice with K-DS and air-dried on a glass slide. Paramecia were incubated in 0.1% of Triton X-100 solution for 30–60 s to increase cell membrane permeability and washed twice with double distilled water (DDW). After blocking with BSA solution (2 mg/mL) for 20 min, paramecia were washed twice with DDW and stained with Alexa488-conjugated NH3545 (anti-immaturin rabbit polyclonal antibody: 6.2 μg/mL, 1/100 dilution by DDW) for 20 min. The specimen was washed twice with DDW and then sealed with VECTASHIELD (Vector laboratories Inc. Berlingam, CA, USA).

### 2.8. Indirect Immunofluorescent Staining

After preincubation for 30 min in K-DS, paramecia were incubated with NH3545 (anti-immaturin rabbit polyclonal antibody: 6.2 μg/mL, 1/100 dilution by DDW) for 1 h, followed by Alexa Fluor 488 F(ab’) goat anti-rabbit antibody (second antibody: 1/200 dilution) for 30 min. Macronuclei were stained with DAPI (4′, 6′-diamidino-2-phenylindole).

### 2.9. Assay for Immaturin-Nuclease Activity

The nuclease activity of immaturin was demonstrated by the agarose gel method. We used purified immaturin fraction (I-fraction) with DEAE-Sephadex ion-exchange column chromatography (1.0 μg protein/lane). DNase I (Sigma, *E. coli*, 10 U/lane) was used as a positive control. Total nucleic acid (TNA) isolated from *Paramecium caudatum* (1.0 μg/lane) was used as a substrate DNA. Each nuclease reaction mixture was incubated at 25 °C for 30 min (A) or 120 min (B, C) at pH 7.0; reactions were stopped by heating at 60 °C for 5 min. The reaction mixture was then cooled using ice before electrophoresis (0.8% agarose (A) or 1.5% (B, C) in TAE at 80 V for 30 min).

Kunitz enzyme assay was performed using ORG590DNase activity (ORGENTEC Diagnostika GmbH). All procedures were performed according to the instructions provided by the manufacturer. Samples were incubated at 25 °C for 60 min for nuclease reaction. The specific activity of immaturin samples was defined as the V_max_ per amount of protein (mg). Data represent an average of two measurements.

### 2.10. GST Assay

The enzyme activity of GST was measured using a GST assay kit (Cayman Chemical Company, Ann Arbor, MI, USA). All procedures were performed according to the instructions provided by the manufacturer. Samples were incubated at 25 °C, and the reaction was initiated by adding CDIXB. Every minute, the reaction mixture was read at 340 nm; this continued for 5 min. V_max_ is represented as an average of the three independent sample measurements.

### 2.11. Paramecium Expression Vector and Transformation

A *P. caudatum* expression vector pTubMcsPcVenus-Immaurin carrying a PcVenus-Immaturin fusion gene was constructed using the pTT3H2B-PcVenus expression vector [19]. pTubMcsPcVenus-Immaurin was made by replacing the H2B (histone H2B gene) site of pTT3H2B-PcVenus with the immaturin gene. After cloning and mass production of this vector, it was linearized with the restriction enzyme BamH1 and used as a sample for microinjection (0.9–1.2 µg/µL). Senescent cells of REM27-1 (*P. caudatum* Syngen 3, mating type O) were cultured on depression slides. Well-fed cells were isolated and approximately 10 pL of the vector was injected into the recipient macronucleus. After standing for about 30 min, the recipients were transferred to a fresh lettuce juice culture medium and incubated at 25 °C. Twenty-four hours after injection, transformants were detected by fluorescence emitted from PcVenus.

## 3. Results

### 3.1. Identification of the Immaturin Gene and Phylogenetic Analysis of the Original Gene

In the first step of cloning the immaturin gene, a soluble immaturin fraction was prepared from cells dividing about 20 times after conjugation [10]. Immaturin was purified as reported earlier [12], and a candidate polypeptide was isolated by SDS-PAGE. Then, partial amino acids of several fragments were sequenced by TOF-MS. One of the identified sequences was INFDGPRLHE. We designed immaturin gene-specific primers (GSPs), matching this sequence in both forward and reverse complement directions: 5′-ATTAACCCATTCGATTCCCAAGA-3′, 5′-TCTTGGGAATCCATCCGAATGGGTTAAT-3′. Total RNA was isolated with an RNA extraction kit (RNeasy, Qiagen, Tokyo, Japan, and 5 ‘and 3’ direction RACE-PCR was performed using the Gene Racer kit (Invitrogen, Waltham, MA, USA).

After completing the immaturin DNA sequence, the C-terminal synthetic peptide QKRLSGGNPYFITL was used as an antibody-producing antigen. This antigenic peptide was injected into a rabbit to obtain a polyclonal antibody called NH3545. To determine whether NH3545 inhibits immaturin activity, the I-supernatant (10.0 mg protein/mL) was used as an immaturin fraction. The immaturin activity of the I-supernatant was inhibited by NH3545 in a dose-dependent manner (Figure 2).

The complete immaturin gene sequence was deposited in GenBank under accession number AGX32176.1. The immaturin gene is predicted to encode 215 amino acids and contains one 26 bp intron. The predicted immaturin protein consists of a glutathione-S-transferase (GST) C family domain and a previously uncharacterized domain (Figure 3).

In order to confirm whether the immaturin gene sequence produces rejuvenation-induced immaturin protein, recombinant immaturin was produced in *E. coli*, and a small amount of recombinant immaturin protein was injected into the cytoplasm of mature Pc cells. Compared to the number of control cells without microinjection, clones derived from microinjected cells had a significantly reduced number of mating-reactive cells (*p* < 0.01). On the other hand, heat-treated (100 °C, 10 min) recombinant immaturin protein had no inhibitory effect on the mating reaction (Figure 4). This implies that the recombinant immaturin protein exhibits immaturin activity and heat-labile chemical properties, similar to the natural immaturin protein [12].

The molecular phylogeny of immaturin is summarized in Figure 5. A BLAST search of ParameciumDB (http://paramecium.cgm.cnrs-gif.fr/cgi/tool/blast accessed on 20 December 2022) [20] using the amino acid sequence of immaturin revealed three, eight, and four immaturin-homologous proteins in *P. caudatum*, *P. tetraurelia* (Pt) and *P. multimicronucleatum* (Pm) (cut-off score > 150 bits), respectively. Of the three Pc homologs found in the database, PCAUDP05425 was the most homologous to immaturin (428 bits), although their protein sequences were not completely identical. Aside from the *Paramecium* immaturin orthologs, a BLAST search of the NCBI protein database identified 17 proteins similar to immaturin (cut-off E value < 1 × 10^−35^), and most were designated as GSTs. These non-paramecium immaturin orthologs were found in α-, β-, γ-proteobacteria, and cyanobacteria.

### 3.2. Immaturin Localization in Subcellular Compartments

Using direct or indirect immunofluorescence and confocal microscopy, the subcellular localization of immaturin was assessed using anti-immaturin antibodies: NH3545 or Alexa488-conjugated NH3545. First, the fluorescence signal of macronuclear immaturin was observed with a confocal fluorescence microscope (Nikon A1, Nikon, Tokyo, Japan) by indirect immunofluorescence. The immature macronuclear immaturin signal was observed as many highly condensed green bodies (Figure 6A). In contrast, the cytoplasmic signal was almost a filamentous structure (Figure 6A), and no visible green body in the mature or senescent macronucleus could be noted.

Secondly, since cilia are organelles that recognize complementary mating types in *Paramecium*, the immaturin localization in cilia was investigated by direct immunofluorescence. After the conjugation process, the progeny cells continued to undergo asexual cell division. Five fissions after conjugation, the cells showed a fluorescent immaturin signal in both cytoplasm and cilia (Figure 6B). The fluorescent immaturin signal was observed continuously throughout the immaturity period. Contrary to expectations, immaturin signals were observed in mature cells with fluorescence intensity comparable to immature cells. However, cilia of senescent cells that did not express mating activity (about 1000 divisions after conjugation) did not display a fluorescence-immaturin signal. In senescent cells, the immaturin signal was detected only on the cell surface.

Finally, we investigated the immaturin signal intensity in the cilia of cells that were transformed with the immaturin gene. The non-transformed group (n = 24) had no percentage of cells exhibiting mating activity, whereas the transformed group had 75.9% (standard deviation: 7.4, n = 51) of cells exhibiting mating activity. As shown in Figure 6B, the cells of transformants were covered with a fluorescent-immaturin signal. The growth rate of transformed cells was significantly increased compared to control senescent cells (*p* < 0.01) (Figure 6C).

### 3.3. Immaturin Digests DNA and Retroviral Genomic RNA

Since the immaturin gene has a GST C-family domain, GST catalytic activity was tested on the supernatant fraction obtained through ultracentrifugation at 105,000× *g* and on the immaturin fraction purified by Sephadex G-50 column chromatography. GST activity was detected in the supernatant, but not in the immaturin fraction (Table 1). These findings indicate that active GST was extracted by ultracentrifugation, and that immaturin protein was separated from GST by using Sephadex G-50 column chromatography.

Based on the macronuclear fusion-organization experiments, immaturin might interact with nuclear chromosomes or chromosomal DNA [15]. The purified immaturin shows strong nuclease activity (Figure 7A). In particular, the immaturin antibody NH3545 inhibited this activity (Figure 7A (lanes 3 and 4)) but did not inhibit *E. coli* DNase I activity (Figure 7A (lanes 6 and 7)). The optimal pH for nuclease activity via immaturin was 6.5–8.5 (data not shown). Immaturin nuclease digested single- and double-stranded DNA (Figure 7B,C), linear and circular DNA, and virus genomic single-stranded RNA (Figure 7C), but did not digest ribosomal RNA isolated from paramecia (data not shown). The V_max_ of immaturin nuclease measured in Kunitz units was higher than DNase I (*E. coli*) (Table 2). Since immaturin single-stranded RNA nuclease activity was not inhibited by 1 mM EDTA, Mg ions were not required. However, immaturin nuclease required Mg ions for DNA digestion, like *E. coli* DNase I (Figure 7C).

## 4. Discussion

The finding that the mating ability of senescent cells is restored by immaturin in microinjection experiments has long been questioned as a contradictory phenomenon. We attempted to answer this question by elucidating the differential gene expression theory that is developed in the eukaryotic cell differentiation process. In this study, we recognized several immaturin gene paralogs in the *Paramecium* genome. In addition, we clarified the intracellular presence of the immaturin peptide by immunofluorescence. These findings have given us new perspectives on the intracellular spatial distribution of immaturin polypeptides.

The microinjection method worked effectively as a bioassay for soluble components. We also clarified the mode of existence of immaturin polypeptides in cilia. It is now possible to have a bird’s eye spatial view of the immaturin world. Hiwatashi reported that cilia growing on the ventral side of the cell body are specifically involved in the mating reaction [2]. We investigated the fluorescence signal of immaturin in the ventral cilia and assessed the correlation between the effects of immaturin and the expression of mating activity. As a result, immaturin signals were detected in the ventral cilia of immature cells. In mature cells, no immaturin signal was detected in ventral cilia. On the other hand, in senescent cells, immaturin signals were detected in cells with restored mating activity. It was suggested that immaturin is involved in the suppression of the mating reaction in ventral cilia in both immature and mature cells. However, the mechanism of enhanced expression in senescent cells remains a mystery.

The nucleotide sequence of the immaturin gene indicates that it originates in the GST cytosol family (GST C family) of proteobacteria. The gene encoding GST is found in all organisms, including prokaryotes and eukaryotes, and the GST proteins are grouped into three superfamilies known as cytosolic, mitochondrial, and microsomal proteins [21,22,23]. The differences in the amino acid sequence of these superfamilies represent complex gene evolution, suggesting their importance in physiological functions, such as detoxification of endogenous compounds and degradation of xenobiotic compounds [24,25,26,27]. *Paramecium* might use the basic structure of the GST protein, dramatically alter its activity, and evolve the immaturin protein with nuclease activity.

Immaturin orthologs are found in the closely related species of *Paramecium* and prokaryotes such as α-, β-, and γ-proteobacteria. In particular, the immaturin orthologs found in *Tetrahymena* (XP001014046.1, 75.5 bits) and *Oxytricha* (EJY85640.1, 53.9 bits) are much less similar to immaturin than prokaryotic orthologs (134-155 bits). A BLAST search using the sequences of *Tetrahymena* and *Oxytricha* immaturin orthologs revealed high-scoring sequences mainly in eukaryotes such as fungi, bonefish, and bivalves. Therefore, we propose that the ancestral immaturin gene could have been first diversified in prokaryotes. Then, *Paramecium* might have received one of the ancestral immaturin genes with a different history than *Tetrahymena* and *Oxytricha*. One possibility is that the ancestral immaturin gene could have been moved horizontally from the bacteria taken by *Paramecium*.

The DNase I enzyme plays a role that includes DNA digestion and chromosomal DNA destruction during drug-induced apoptosis [28,29]. DNase I is used as a probe for protein–DNA interactions in the laboratory, such as DNase hypersensitivity assays and foot-printing techniques to identify transcription factor binding sites [30]. In addition, recent advances in DNase I enzyme research indicate that the DNase I enzyme exhibits some sequence-specific digestive activity [31]. The opposite effect of immaturin observed in the laboratory life-cycle phase transitions may be due to different DNA-binding proteins in mature and senescent cells. In mammalian cells, DNase I is implicated in chromatin remodeling [32] and B cell function in SLE (systemic lupus erythematosus) [33]. The general characteristics of several types of endonucleases have already been summarized [34].

In conclusion, this study reveals that immaturin polypeptides, originally thought to exist only in immature cells, have been found to exist in all the stages of life history (Figure 8). Immaturin was found in the macronucleus, cytoplasm, cell surface area, and cilia during the immaturity period and only in the cell surface area during the senescence period (Figure 6B). The transition of the existence of immaturin mode is thought to cause sexual development and aging. The rejuvenation from the age of senescence due to the introduction of the immaturin gene suggests that the immaturin gene itself is the main cause of aging. As a general method for artificially induced rejuvenation, we propose identifying genes that function continuously from early to late life and providing treatments that complement their functional decline. Advances in induced pluripotent stem cell (iPS) technology in mammalian organisms demonstrate that transcription factors are important in inducing the reprogramming of differentiated cells [35,36]. The discovery of the *Paramecium* immaturin nuclease opens new avenues for studying the life-cycle phase transitions mediated by nucleases. The immaturin gene may function as an intracellular nuclease that effectively digests the genome of invading viruses. In addition, since the basic conditions for mass production of the immaturin gene and immaturin polypeptide in *E. coli* cells have been established, it is attractive as a useful tool for the medical treatment of viral infections.

## Figures and Tables

**Figure 1 microorganisms-11-00082-f001:**
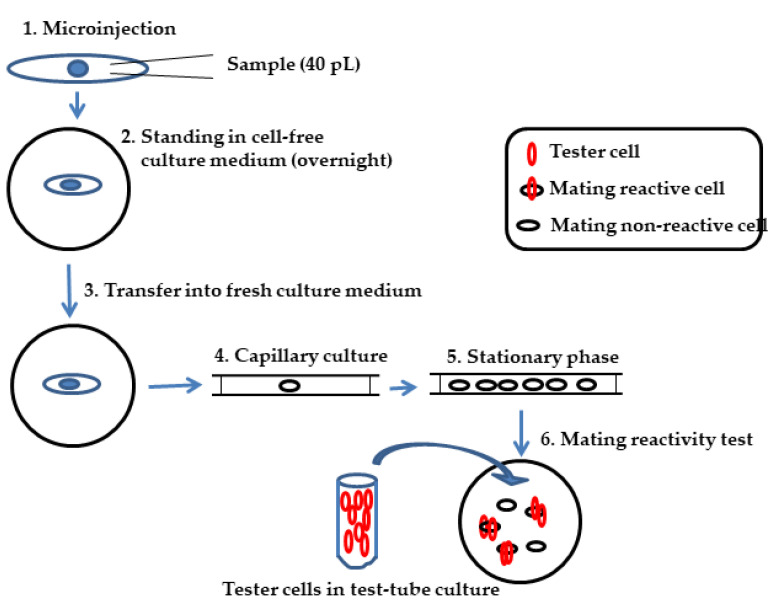
Schematic of immaturin bioassay with microinjection and capillary culture. Since the odd-mating-type cells were used as the recipients of microinjection, the even-mating-type cells were used as the tester to evaluate the mating activity. The strength of the mating activity of each clone derived from microinjection is expressed as the percentage of cells mated with the tester to the total number of cells used in the test. The effect of injection of immaturin was determined to be when less than 70% of the cells constituting the clone had a mating reaction.

**Figure 2 microorganisms-11-00082-f002:**
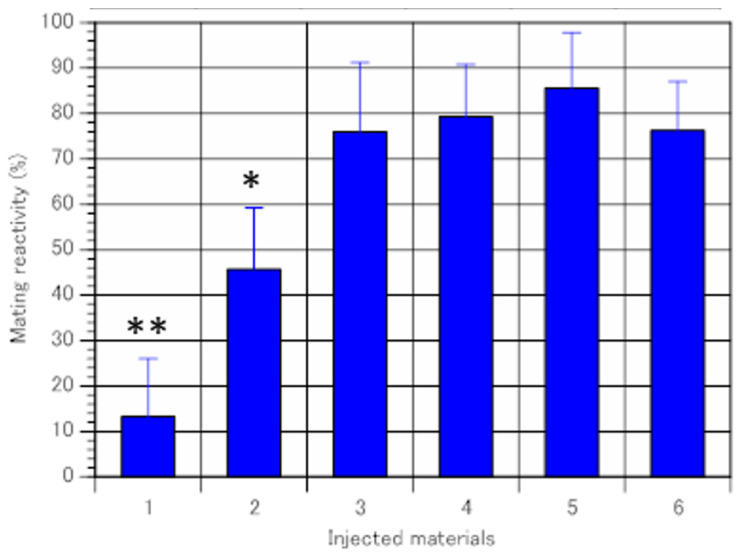
Inhibition of immaturin activity by the anti-immaturin antibody. The vertical axis indicates the percentage of mating reactive cells in the clones derived from recipients. The horizontal axis represents the injected samples. 1. Immaturin, 2. Immaturin mixed with anti-immaturin antibody (0.15 mg/mL), 3. Immaturin mixed with anti-immaturin antibody (0.31 mg/mL), 4. anti-immaturin antibody (0.31mg/mL), 5. Buffer solution, 6. Without injection. Data represent the average percentage of mating reactive cells ± SD (n = 6). Tukey’s multiple comparison test: * *p* < 0.05, ** *p* < 0.01.

**Figure 3 microorganisms-11-00082-f003:**
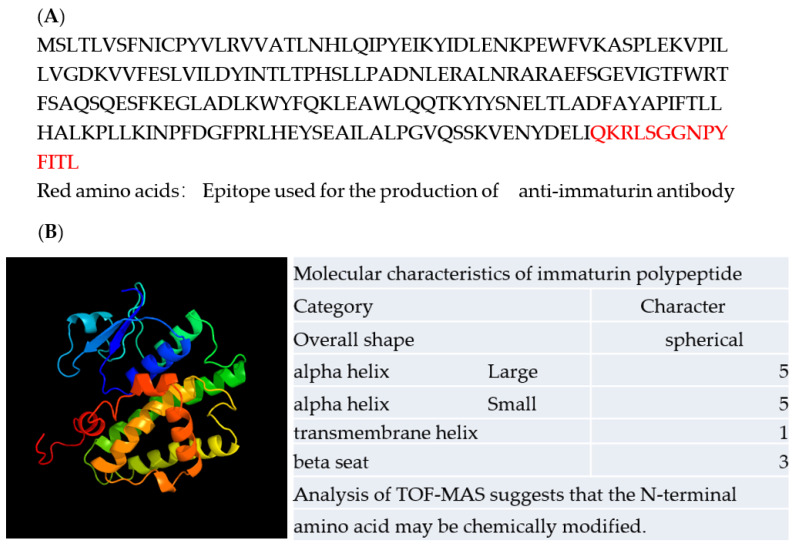
A deduced amino acid sequence of immaturin and molecular information. (**A**) The amino acid sequence QKRLSGGNPYFITL was used as an epitope for the production of the anti-immaturin antibody NH3545. (**B**) A three-dimensional ribbon model of immaturin is displayed (left panel). The N-terminus of the immaturin polypeptide is shown in blue and the C-terminus in red. The ribbon model was produced by the computer graphic software Phyre 2 (protein homology/analogy recognition engine v. 2.0). The total shape deduced from the amino acid sequence of immaturin and the types and numbers of primary structures are shown in the table (right panel).

**Figure 4 microorganisms-11-00082-f004:**
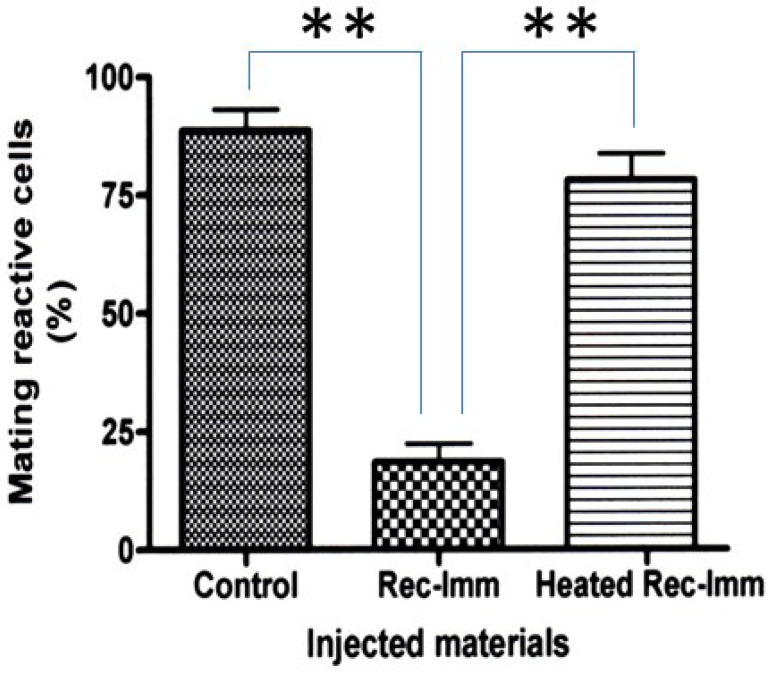
Recombinant immaturin produced by *E. coli* cells inhibits the expression of mating reactivity in mature *Paramecium* cells. Microinjection assay was performed as described in Materials and Methods. Rec-Imm indicates recombinant immaturin. Heated Rec-Imm indicates rec-immaturin treated at 100 °C for 10 min. Data represent the average percentage of mating reactive cells ± SD (n = 6). Tukey’s multiple comparison test: ** *p* < 0.01.

**Figure 5 microorganisms-11-00082-f005:**
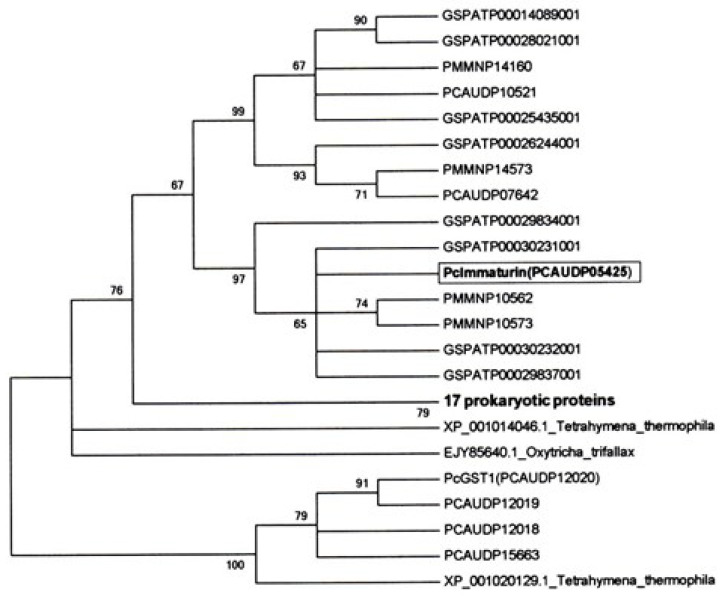
The maximum likelihood (ML) consensus tree of immaturin. A phylogenetic tree was generated from bootstrap analysis with 1000 replications of immaturin and its orthologs based on amino acid sequence alignment shown in Appendix A. Species names are indicated with the protein IDs used in the Paramecium DB or GenBank accession numbers. Pc, Pt, Pm, Tt, and Ot are abbreviations of *Paramecium caudatum*, *P. tetraurelia*, *P. multimicronucleatum*, *Tetrahymena thermophile,* and *Oxtricha trifallax*, respectively. Bootstrap values > 60% are given to the left of the selected nodes. Branches with less than 50% of bootstrap replicates are collapsed. Seventeen prokaryotic orthologs are compressed and shown as a single node.

**Figure 6 microorganisms-11-00082-f006:**
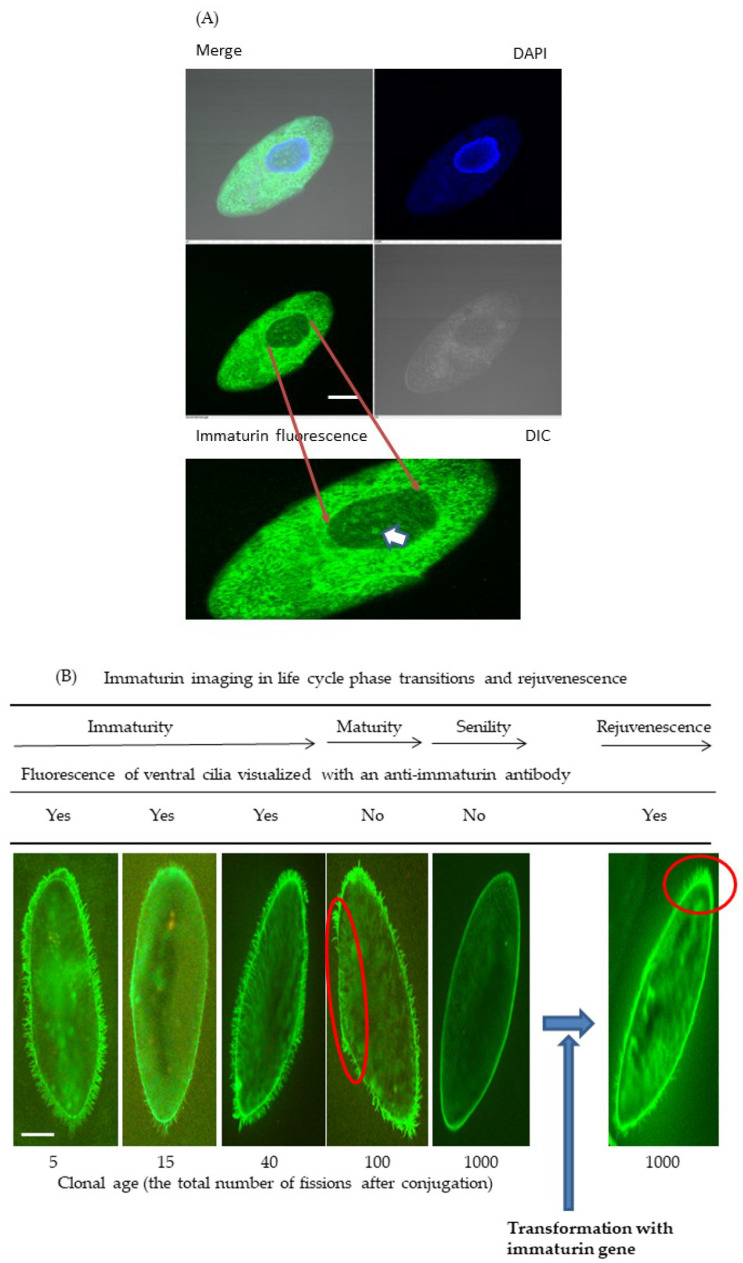
Immaturin-imaging in the life cycle phase transitions and rejuvenescence. (**A**) Confocal images of immature cells at about 20 fissions after conjugation, photographed using Nikon A1. Representative cells show many condensed green bodies (GB) in the macronucleus (an arrowhead indicates one typical GB). Scale bars, 20 μm. (**B**) The early stage of immaturity at five fissions after conjugation. The middle stage of immaturity is at about 15 fissions after conjugation. Late-stage immaturity is at about 40 fissions after conjugation. The mature stage is at about 100 fissions after conjugation. The senescent stage is at about 1000 fissions after conjugation. Senescent cells at about 1000 fissions were transformed by the microinjection of the immaturin gene. The ‘yes’ or ‘no’ attached to each stage of the picture visualized with an anti-immaturin antibody indicates the presence or absence of fluorescence in the ventral cilia, respectively. Scale bars, 20 μm. (**C**) The graph shows the rejuvenating effect of immaturin gene transformation on the proliferative capacity of senescent cells. Single cells were isolated in a capillary with the fresh culture medium in each experimental group and incubated for four days at 25 °C. Old cont. indicates control senescent cells. I gene-Old indicates senescent cells transformed by the immaturin gene. Data represent average total cell numbers in the capillary culture ± SD (n = 6). Student’s t-test: ** *p* < 0.01.

**Figure 7 microorganisms-11-00082-f007:**
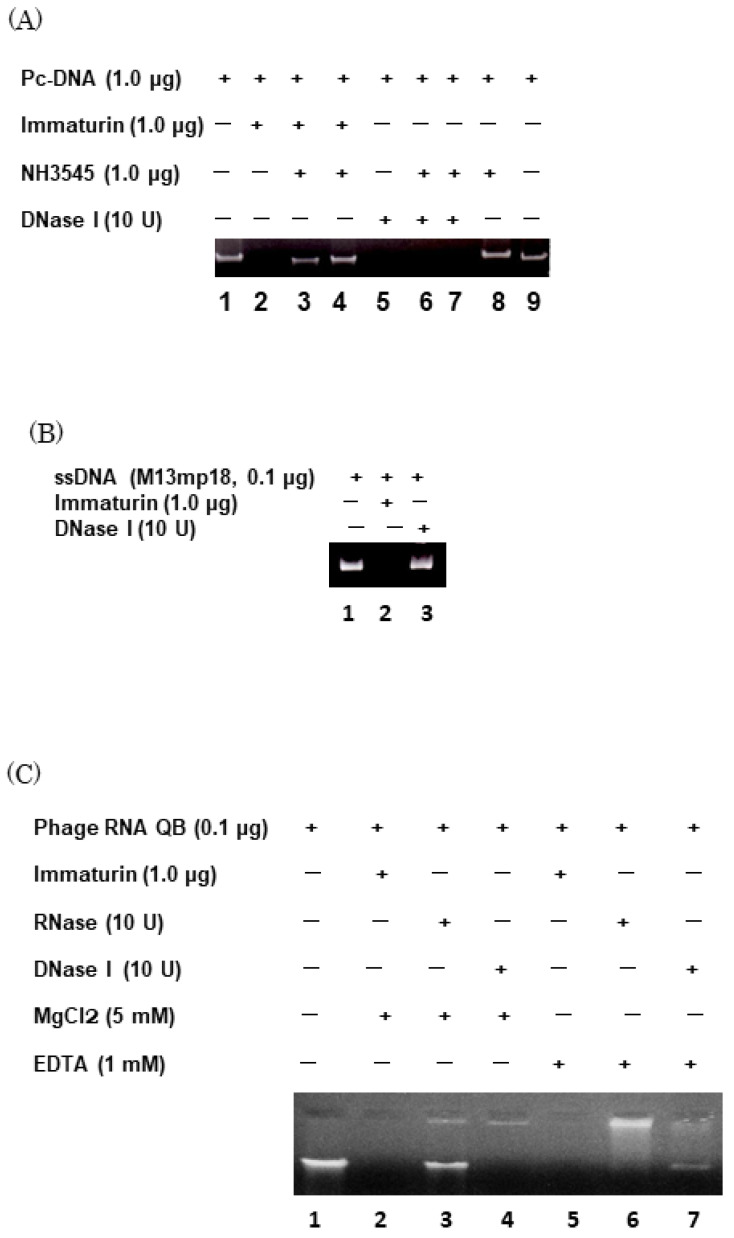
Characterization of immaturin-nuclease activity. (**A**) The purified immaturin fraction was pre-incubated with NH3545 (0.15 (lanes 3 and 6) or 0.31 mg protein/mL (lanes 4, 7, and 8)) for 24 h at 4 °C. Immaturin-antibody NH3545 inhibited immaturin nuclease activity, but *E. coli* DNase I was not inhibited by NH3545. (**B**) Immaturin nuclease digested single-stranded DNA, but *E. coli* DNase I did not. (**C**) Immaturin nuclease digested double-stranded phage genomic RNA (QB), but RNase A (*E. coli*) did not.

**Figure 8 microorganisms-11-00082-f008:**
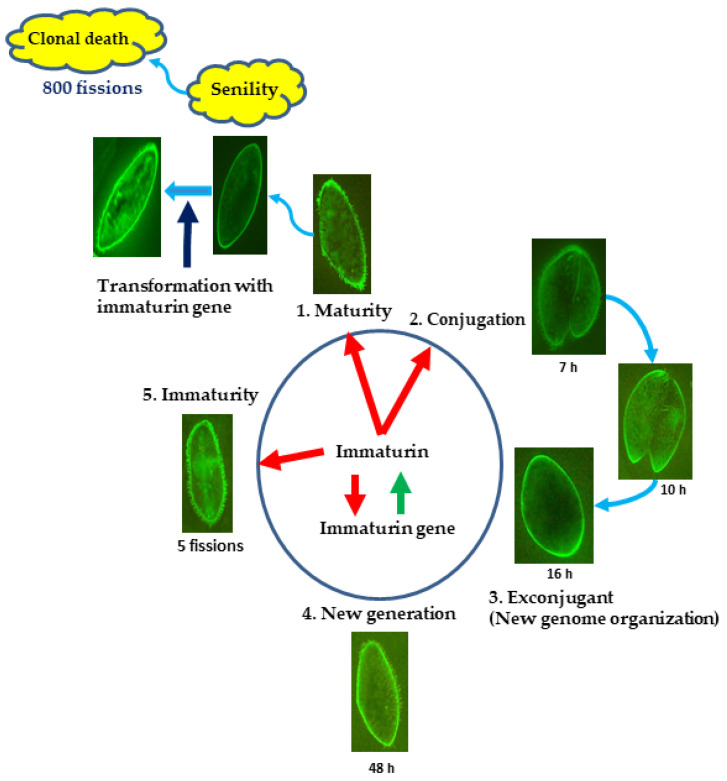
Schematic diagram of the action of immaturin gene and immaturin polypeptide. Photographs in the schematic diagram show fluorescence images of immaturin at each stage of the life cycle. The formation of a new generation begins with the mating reaction of mature cells. In the mating reaction, two cells adhere to form a fertilization nucleus with their own germ nuclei. The new generation begins in the immaturity stage, passes through the maturity stage, and shifts to the senile stage. Our investigation detected immaturin at all life cycle stages. In microinjection experiments, immaturin suppresses mating ability during the immaturity stage and promotes mating ability during senescence. The results of visualization experiments by immunofluorescence provide molecular clues for understanding the contradictions in the microinjection experiments. The red arrow indicates the direction of action of immaturin and the green arrow indicates the synthesis of immaturin polypeptides from the immaturin gene.

**Table 1 microorganisms-11-00082-t001:** Glutathione-S-transferase enzyme assay of immaturin.

Samples	Protein (mg/mL)	GST Activity (V_max_ (mU/min))
105,000× *g* supernatant	2.30	18.34
Sephadex G-50 Immaturin fraction	0.60	0.70
DEAE-Sephadex A25 Immaturin fraction	0.43	0.00

These immaturin fractions were prepared by the previously reported method [13].

**Table 2 microorganisms-11-00082-t002:** DNase I enzyme assay of immaturin.

	Protein (mg/mL)	Specific Activity
		(V_max_ (mKu/mL)/Protein)
105,000× *g* supernatant	2.30	9.6
Sephadex G-50 Immaturin fraction	0.60	26.7
DEAE-Sephadex A25 Immaturin fraction	0.43	11.6
DNase I (*E. coli*)	1.00	4.0

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
