# Peer review of "Immaturin-Nuclease as a Model System for a Gene-Programmed Sexual Development and Rejuvenescence in Paramecium Life History"

_microorganisms, 2022, doi:10.3390/microorganisms11010082_

Round 1
Reviewer 1 Report
I found this passage hard to. understand, how many were found for which species? "of ParameciumDB (http://paramecium.cgm.cnrs-gif.fr/cgi/tool/blast) [16] using the amino 213 acid sequence of immaturin: 3 for P. caudatum, 8 for P. tetraurelia (Pt), and for P. 214 multimicronucleatum (Pm), 4 immaturin homologous proteins was revealed (cut-off score>"
In the introduction, the nature of inmaturity isn't introduced before discussed : . I"n most species of ciliates, the length of the 34 immaturity period is determined"
No triangle: "Seventeen prokaryotic orthologs are 229 compressed and shown as a closed triangle."
In Figure 6B, I don't understand the noted lack of staining in the ventral cilia in the 100 fission cell. It looks positive to me—perhaps a higher magnification of these structures would help? The almost ubiquitous staining is what most worries me in the paper.
Author Response
Response to the first reviewer's comments
- We have corrected the description.
P12, line 212
The molecular phylogeny of immaturin is summarized in Figure 5. A BLAST search of ParameciumDB (http://paramecium.cgm.cnrs-gif.fr/cgi/tool/blast) [16] using the amino acid sequence of immaturin revealed 3, 8, and 4 immaturin-homologous proteins in P. caudatum, P. tetraurelia (Pt) and P. multimicronucleatum (Pm) (cut-off score> 150 bits), respectively. Of the 3 Pc homologs found in the database, PCAUDP05425 was the most homologous to immaturin (428 bits), although their protein sequences were not completely identical. Aside from the Paramecium immaturin orthologs, a BLAST search of the NCBI protein database identified 17 proteins similar to immaturin (cut-off E value <1e-35), and most were designated as GST. These non-paramecium immaturin orthologs were found in α-, β-, γ-proteobacteria, and cyanobacteria.
P13, line 229
are compressed and shown as a single node.
- We described the findings on the nature of immaturity in other ciliates on page 1 in lines 34-36 of the introduction.
- We have corrected the description. P13, line 229
- We agree with your comments.
We think that higher-magnification photomicrographs are needed to determine the presence or absence of the immaturin signal in the ventral cilia.
We faced a difficult problem in detecting the fluorescence signal of cilia that express mating activity. When the ventral cilia were negative and the surrounding cilia were positive, it was very difficult to image the negative cilia. This is because cilia without fluorescent material are too dark to be imaged by fluorescence microscopy.
We did the following experiments. The cells that initiated the mating reaction were isolated and observed within 10 min by immunofluorescence. No immaturin fluorescence signal was detected from the cilia of the cells undergoing mating reaction (unpublished data).

Reviewer 2 Report
The article " Immaturin-nuclease as a model system for a gene‐programmed sexual development and rejuvenescence in Paramecium life history" is devoted to the further study of the role of the immaturin protein in regulating the number of cell divisions, development and aging of Paramecium. In ciliates, a clade of microbial eukaryotes that is estimated to be about 1 billion years old, germline and somatic functions are isolated into distinct nuclei within a single cell/individual. As in animals, the germline remains quiescent throughout much of a ciliate’s life, only becoming transcriptionally active during conjugation. So, Paramecium are a perfect model system for the study of factors regulating the course of the clonal cycle. It was previously shown that immaturin inhibits the mating activity of mature cells.
The authors of the article cloned the immaturin gene, expressed it in E. coli, studied the enzymatic properties of the recombinant protein and showed that it has specific nuclease activity. Using immunofluorescence microscopy, the authors made a logical conclusion that the function of immaturin is determined not only by its presence, but also by its intracellular localization.
The presented article will undoubtedly contribute to further research on the life-cycle regulation in ciliates.
Some remarks and questions:
1. The authors note (Fig. 3B) that the N-terminal amino acids can be modified. Is it known what modifications are taking place?
2. Does immaturin differ in immature and senescent cells?
3. How immaturin moves in the cell (macronucleus, cytoplasm, cilia)? Are there any carrier proteins?
4. Lines 265-266: “However, cilia of senescent cells … did not display a fluorescence-immaturin signal”. What happens with immaturin in cilia? Is it degraded or modified?
5. “Senescent cells at about 1,000 fissions were transformed by the microinjection of the immaturin gene” (fig.6, line 249). Did you really transform Paramecium with the immaturin gene? How did you do it, what vector did you use? What are the conditions for transformation? There is no mention of this in the Methods section.
6. Is there anything known about the immaturin gene copy numbers in the macronucleus of immature and senescent cells? If there are differences, what are they caused by?
The article can be accepted for publication with minor corrections and answers to questions.

Author Response
Response to the second reviewer’s comments
- According to the results from the company that requested the TOF-MAS analysis, it is highly likely that it is phosphorylated.
- There is no molecular information comparing immature and senescent immaturin. The reason is that it is difficult to culture large amounts of senescent cells.
- We have no information on how the immaturin molecule moves within the cell.
- No information is available to explain the lack of immaturin signal in the cilia of senescent cells.
Presuming from the fluorescence intensity, the amount of immaturin in senescent cells is considered to be significantly lower than that in mature cells. There are at least two possible explanations. One is due to a decrease in the number of immaturin genes in the macronucleus and the other is due to a decreased function of the transcription/translation system.
- We have added the following paragraphs to Materials and Methods
Paramecium expression vector and transformation
A P. caudatum expression vector pTubMcsPcVenus-Immaurin carrying a PcVenus-Immaturin fusion gene was constructed using the pTT3H2B-PcVenus expression vector [ ]. pTubMcsPcVenus-Immaurin was made by replacing the H2B (histone H2B gene) site of pTT3H2B-PcVenus with the immaturin gene. After cloning and mass production of this vector, it was linearized with the restriction enzyme BamH1 and used as a sample for microinjection (0.9-1.2 µg/µL). Senescent cells of REM27-1 (P. caudatum Syngen 3, mating type O) were cultured on depression slides. Well-fed cells were isolated and approximately 10 pL of the vector was injected into the recipient macronucleus. After standing for about 30 minutes, the recipients were transferred to a fresh lettuce juice culture medium and incubated at 25°C. Twenty-four hours after injection, transformants were detected by fluorescence emitted from PcVenus.
- There are no experimental results to answer the reviewer's question.
To our knowledge, there are no direct measurements of the number of immaturin genes in the macronucleus.
We think that designing primers that distinguish the paralogous genes of the immaturin gene will provide information for estimating the number of transcriptionally active immaturin genes.
We believe that this experiment is one of the most important projects in testing the genetic programming hypothesis of clonal aging in Paramecium.
